# The impact and cost of a new rapid diagnostic test for school-based prevalence mapping and monitoring and evaluation surveys of schistosomiasis: A modelling study

Joshua M. Chevalier[1], Kyra H. Grantz[1,2], Sarah Girdwood[2], Stella Kepha[3], Thierry Ramos[2], Brooke E. Nichols[1,4], Shaukat Khan[2]*, Sarah Hingel[2]*

1 Department of Global Health, Amsterdam Institute for Global Health and Development, Amsterdam UMC, University of Amsterdam, Amsterdam, Netherlands, 2 FIND, Geneva, Switzerland, 3 Eastern and Southern Africa Centre of International Parasite Control, Kenya Medical Research Institute, Kisumu, Kenya, 4 Department of Global Health, Boston University School of Public Health, Boston, Massachusetts, United States of America

* sakhan@gmail.com (SK); sarah.hingel@finddx.org (SH)

## Abstract

### Background

In endemic communities where the prevalence of Schistosomiasis is ≥ 10%, annual preventive chemotherapy is recommended. Traditional sampling methods determine infection prevalence through district-level surveys in school-aged-children (SAC). Recently, an alternative sampling strategy—the Schistosomiasis Practical and Precision Assessment (SPPA) protocol—was developed to aid in targeting treatment to the sub-district level. A prototype circulating anodic antigen rapid diagnostic test (CAA RDT) could avoid the pitfalls associated with current microscopy techniques and therefore be better suited to support precision-mapping.

### Methodology

We modelled the ability of a CAA RDT to correctly classify sub-district prevalence above or below the 10% threshold in simulated districts under alternative sampling strategies. Each district (10 sub-districts/district) had varying mean prevalence and prevalence distributions. Test sensitivity (60–100%) and specificity (95–100%) of the CAA RDT was varied. We then determined the associated survey costs for prevalence mapping or monitoring and evaluation for each sampling strategy using the CAA RDT compared to microscopy.

### Results

The CAA RDT cost/SAC was US$12.14, which was similar to Kato-Katz (US$13.23/SAC) using traditional sampling. Sampling with the CAA RDT cost the least when

**Data availability statement:** The data that supports the findings in this study is publicly available at: https://doi.org/10.6084/m9.figshare.c.7515291.v2.

**Funding:** Funding for this study was awarded to FIND by Ares Trading S.A., a division of Merck (https://www.merckgroup.com/ch-en). The funders had no role in the study design, data collection and analysis, decision to publish or preparation of the manuscript.

**Competing interests:** The authors have declared that no competing interests exist.

conducting SPPA sampling or M&E, or when both Kato-Katz and urine filtration were required. High specificity of the CAA RDT was a key determinant of performance and a test with 100% specificity and 85% sensitivity correctly classified the most sub-districts (87%) under SPPA sampling. SPPA sampling generally led to less under- and overtreatment of sub-districts compared to traditional sampling.

## Conclusions

A CAA RDT with high specificity will lead to similar treatment success at lower costs, under either sampling strategy, as compared to Kato-Katz and urine filtration. The CAA RDT could be a valuable diagnostic tool for determining schistosomiasis preva-lence and could better support precision mapping strategies through reduced costs, thereby improving mass drug administration and aiding programmes to eliminate schistosomiasis as a public health problem.

## Author summary

Control of schistosomiasis centers around regular district-wide preventive treatment based on prevalence surveys of the district using standard micros-copy methods. As schistosomiasis is a focal disease, these broad prevalence surveys can lead to over- or undertreatment, particularly of communities with a prevalence close to the 10% threshold that indicates mass-treatment. Precision mapping surveys that target treatment to the sub-district level could refine the use of mass treatment, providing for more efficient use of resources. Using a mathematical model, we compared the impact and cost of a novel rapid diagnos-tic test for schistosomiasis to that of standard microscopy and evaluated its use in a precision mapping strategy. We show that a rapid diagnostic test with com-parable performance metrics to microscopy would be cost saving when used in a precision mapping approach and could lead to less over- and undertreatment at the sub-district level compared to traditional surveys.

## Introduction

The World Health Organization's (WHO) roadmap for neglected tropical diseases has set a target to eliminate schistosomiasis (SCH) as a public health problem in 78 countries by 2030 [1]. This is defined as less than 1% of all infections in a region being of heavy intensity (≥ 50 eggs per milliliter of urine or ≥ 400 eggs per gram of stool) [2]. In endemic communities with a SCH prevalence of ≥ 10%, WHO guide-lines recommend annual preventive chemotherapy (using a single dose of prazi-quantel) for all age groups from two years old and older to control SCH [3]. The prevalence is estimated by parasitological microscopy upon stool examination with Kato-Katz smears for intestinal SCH (species *S. mansoni/ S. japonicum/ S. mekongi/ S.guineensis/ S.intercalatum*), and urine filtration with microscopy for urogenital SCH

(*S. haematobium*). New guidelines lowering the prevalence threshold to 10% for annual preventive chemotherapy and extending treatment to adults will necessitate a larger global supply of praziquantel which could exceed currently available stocks via donation schemes [4]. Strategies are needed to support more prudent, targeted use of available treatments to reduce disease burden and achieve SCH elimination targets.

Currently, SCH prevalence is determined through district-level, school-based mapping surveys in school-aged children (SAC), as they are deemed to be an appropriate proxy of community-level infection. SCH is a focal disease, often characterized by a heterogeneous distribution of prevalence and infection intensity, even within smaller geographic regions (i.e., between communities) [5,6]. As such, precision mapping of SCH at lower administration units (sub-district level) could allow for spatially targeted treatment of at-risk communities and thus reduce the overall need for praziquantel [7,8]. Recently, the Schistosomiasis Oversampling Study (SOS) recommended an optimal two-stage sampling method—termed the Schistosomiasis Practical and Precision Assessment (SPPA) protocol—to ensure efficient, appropriate treatment classifications for sub-districts above or below the 10% threshold [9]. The first stage of sampling is at the district-level to identify if the district has a homogeneous or heterogeneous SCH prevalence distribution. If district prevalence is deemed heterogeneous, additional sampling is required at a sub-district level for a sub-district level treatment decision. The frequency and scale of impact assessments are hampered by the time- and labor-intensive nature of microscopy-based diagnostics. Microscopy requires obtaining stool and urine samples—over multiple days in the case of monitoring and evaluation (M&E) surveys—transport to a laboratory, and trained personnel.

A circulating cathodic antigen (CCA) rapid diagnostic test (RDT) has been commercially available as a point-of-care test since 2008, however, despite being easier to use and less labor intensive than Kato-Katz, it only works well for detecting *S. mansoni* infections and its sensitivity decreases with decreasing infection intensity [10]. A new RDT (still under development) can detect circulating anodic antigen (CAA) in an infected host's blood, as a marker for active infection for *S.mansoni, S.japonicum* and *S.haematobium* [11]. Based on early evaluations, the CAA RDT prototype has the potential to reliably measure prevalence in SAC as well as in adults [12,13]. The WHO's diagnostic target product profile (TPP) for a SCH RDT has recommended a minimal sensitivity of > 60% and a specificity of > 95% for *S.mansoni* and/or *S.haematobium* [14]. The CAA RDT prototype is under field testing and is expected to meet and potentially exceed the WHO's TPP recommendation and thus would be a valuable tool for precision mapping of SCH in endemic communities. With its availability as a point-of-care RDT, it enables more widespread testing within the general population, which can inform targeted interventions and support mass drug administration (MDA) of praziquantel as well as measure the impact of the interventions. By identifying communities with SCH, the CAA RDT could help guide the allocation of resources and interventions to where they are needed most.

In this analysis we first aimed to determine the overall operational costs of using the CAA RDT compared to microscopy-based methods (Kato-Katz, urine filtration) through traditional or SPPA sampling strategies. We then modelled the ability of the alternative sampling strategies to estimate prevalence, with the respective diagnostic technologies, in mathematically simulated districts with various underlying prevalence distributions to determine the relative treatment outcomes. This analysis seeks to provide evidence on the interplay between costs, test performance, and sampling accuracy to guide test adoption decisions, especially given recommendations for new sampling strategies.

## Methods

### Diagnostic technologies

We modelled the following diagnostic testing strategies used in prevalence mapping and monitoring:

(1) *Kato-Katz and urine filtration*

Under the current standard testing algorithms, selected SAC provide a stool sample, collected at school, for microscopic evaluation for the presence of *S. mansoni* or *S. japonicum* infection in a nearby laboratory (Kato-Katz technique).

In areas with known circulation of *S. haematobium*, urine filtration is conducted with or without Kato-Katz evaluation of stool samples. One urine sample per SAC is collected at the school, which is first evaluated with haemastix and then further evaluated by a laboratory technician via microscopy in a nearby laboratory. For M&E purposes, stool and/or urine samples are collected for each of the SAC on two consecutive days and evaluated via microscopy in a nearby laboratory.

(2)  *CAA RDT*

With implementation of the CAA RDT (DCNDx, USA), selected SAC are evaluated for *S. mansoni*, *S. japonicum* and *S. haematobium* infection using a single finger-prick capillary whole blood sample collected by a laboratory technician and read at the school by a field assistant.

## Cost data and resource use

We used an ingredients-based approach to identify and quantify all the inputs required to perform the respective test, as well as their estimated quantities and prices [15]. Costs were included from the provider-perspective—in this case the respective National SCH Programme. Resource use was determined through interviews with individuals involved in the implementation of the first field evaluation of the SCH CAA RDT prototype in Kenya by the Kenya Medical Research Institute (KEMRI). Key input categories included labour, transport, supplies, equipment, training, capital and overhead costs (Table A in S1 Text). The costs included all expenses related to specimen collection, sample processing and analysis, data management and treatment delivery, following the approach of Worrell et al [4]. Where available, actual prices were applied, while budgeted prices were used in cases where actual expenditure data was not accessible. In this model, the costs associated with treatment were not included, as the focus was on understanding the potential cost savings associated with the use of the CAA RDT relative to current testing algorithms. All costs were inflated to 2023 USD, and where applicable, converted to USD from Kenyan Shillings (KS) using an exchange rate of 154 KS/USD. All costing workbooks and sensitivity analyses are available in the S1 Text (Table A and Fig A in S1 Text). To ensure the costs are generalizable to other regions and countries (e.g., the Philippines for *S.japonicum*), which may differ in terms of geographic accessibility and staff salaries, a sensitivity analysis using lower staff and transport costs was conducted to represent smaller, more densely populated districts.

## Sampling strategies

We modelled two SCH sampling strategies with each diagnostic technique: the traditional sampling strategy previously outlined by the WHO ("traditional sampling") and the two-stage SPPA method developed through the Schistosomiasis Oversampling Study ("SPPA sampling") [9,16,17]. For each of the sampling strategies, we also included two consecutive days of sampling for microscopy to approximate M&E impact evaluations.

(1)  The traditional sampling method selects 5 schools throughout a district and tests 50 SAC per school [16,18]. Sites are selected purposively, meaning sites with a higher potential for SCH (i.e., proximity to water bodies or known sites of transmission) are selected preferentially as outlined in the WHO guidelines. As a proxy for purposive sampling in the model, we used a sampling weight proportional to the maximum school prevalence in the simulated district (site prevalence divided by maximum school prevalence). After sampling, the average prevalence across the 5 sites was used as the decision metric for treatment (increasing MDA frequency if above 10% prevalence; decreasing or maintaining MDA frequency if below 10%) at the district level.

(2)  In the SPPA sampling strategy, the first phase, *practical assessment*, is meant to determine whether prevalence is homogeneously or heterogeneously distributed in a district. In this phase, 15 schools are systematically selected per district to ensure equal coverage of sites across sub-districts (meaning at least one school is selected per sub-district) and 30 SAC are sampled per school. The number of sites above 10% prevalence inform whether MDA frequency is

decreased or kept constant for the entire district (0–1 sites with prevalence ≥ 10%), whether MDA frequency in the district is increased (8 or more sites with prevalence ≥ 10%), or whether phase 2 of the sampling scheme, *precision assessment,* is indicated (2–7 sites with prevalence ≥ 10%) [9,17]. In precision assessment, four schools are purposively selected from each sub-district and 20 SAC are tested for SCH. Decisions to increase or maintain/decrease MDA frequency are made for each sub-district, as determined by whether the average sub-district prevalence across all sites is above or below the 10% threshold. In practice, if it is known *a priori* that a district has a heterogeneous prevalence distribution, practical assessment may be skipped to proceed directly to precision assessment. We have ignored this possibility in our modelling.

### District and diagnostic sampling simulation

We simulated eight different district archetypes, each consisting of 10 sub-districts and 50 schools per sub-district, with varying prevalence distributions. District size, with respect to the number of sub-districts and primary schools per sub-district, was parameterized to demographic data from Kenya [19–21]. We assumed an average district size of one-million people, divided into 10 equal sub-districts of 100,000 people [19,20]. We assumed 20% of the population (20,000/sub-district) are school-aged children, divided into schools of an average size of 400 students [21]. District and sub-district refer to the county and sub-county administration units (levels two and three, respectively) of the country. District archetypes included a low prevalence district (average prevalence 5%); moderate prevalence districts near the 10% prevalence threshold (8% and 12%), and a high prevalence district (20%), all with either a homogeneous or heterogeneous distribution of sub-district prevalence (Table 1). Each district archetype is simulated 1000 times, with sub-district prevalence randomly drawn from a beta distribution with specified mean SCH prevalence and a corresponding 95% distribution interval target informed by prevalence studies conducted in Kenya (Table 1) [22–26]. Prevalence among children at each school within a sub-district is assumed to be homogeneously distributed (standard deviation of 0.75% around the mean sub-district prevalence). As a sensitivity analysis, we modelled three additional district archetypes with bimodal prevalence distributions—an 8% prevalence district with seven sub-districts below (5%) and three sub-districts above (15%) the 10% prevalence threshold, a 10% prevalence district with five sub-districts below and five above 10%, and a 12% prevalence district with three sub-districts below and seven above 10%.

The sampling strategies (traditional, SPPA) and the diagnostic technologies (specified via test sensitivity and specificity) were modelled over each of the 1000 iterations of the simulated district archetypes. The estimated average prevalence of the district (traditional sampling, SPPA phase 1) or of the sub-districts (SPPA phase 2) determined what the treatment outcome would be for the sub-districts in relation to the 10% prevalence threshold. The performance and cost of the CAA RDT was compared to Kato-Katz (*S. mansoni, S. japonicum*) or urine filtration (*S. haematobium*) across all district archetypes and sampling strategies for different levels of CAA sensitivity (60–100%) and specificity (95–100%), the lower bounds of which were informed by the minimum requirements identified in the WHO TPP [14]. Kato-Katz and urine filtration were assumed to be 80% sensitive and 100% specific for one sample read over two slides [27,28]. The mean, median, 95% confidence interval, and interquartile range were calculated across the simulations for all outcome measures. Modelling and analysis were conducted in R version 4.3.2.

**Table 1. District archetype average and 95% confidence interval of sub-district prevalence.**

| Mean Prevalence | Homogeneous 95% CI | Heterogeneous 95% CI |
| --- | --- | --- |
| 5% | 3.7–6.6% | 0–15% |
| 8% | 6.5–9.6% | 1–20% |
| 12% | 10.5–13.5% | 2–26% |
| 20% | 18.5–21.5% | 3–49% |

## Outcomes

The outcomes of interest in this analysis were the total cost of the respective testing algorithms and the proportion of sub-districts in each sampled district that would receive the correct treatment decision as determined through the modelled simulations (i.e., increasing frequency of praziquantel MDA when true sub-district prevalence was ≥ 10% or decreasing or maintaining MDA frequency if prevalence was < 10%). We sought to assess the test specifications and sampling strategy that could classify at least 80% of sub-districts with the correct treatment decision.

## Results

### Costs of testing strategies

The cost of testing a district using traditional sampling with the CAA RDT was slightly less than the cost of testing using Kato-Katz (Fig 1). The principal cost drivers were supplies (46%), primarily due to the assumed cost of the test kit ($3), and transport (21%) for the CAA RDT prototype, versus personnel (34%) and transport (27%) for Kato-Katz reflecting the additional time required by staff in the laboratory and to return results to the school. Testing with urine filtration alone cost less than the CAA RDT but was considerably more expensive when combined with Kato-Katz microscopy. When conducting M&E of a district, where two samples are required per SAC, the cost of testing with Kato-Katz was more than twice the cost of the CAA RDT.

   The total costs of mapping using SPPA phase 1 or phase 1 and 2 sampling, as well as the cost per SAC are presented in Fig 2. Kato-Katz was always more expensive than the CAA RDT per SAC across all sampling strategies. SPPA sampling for phase 1 was only 8–20% more expensive per SAC than traditional sampling across all diagnostics, but if phase 2

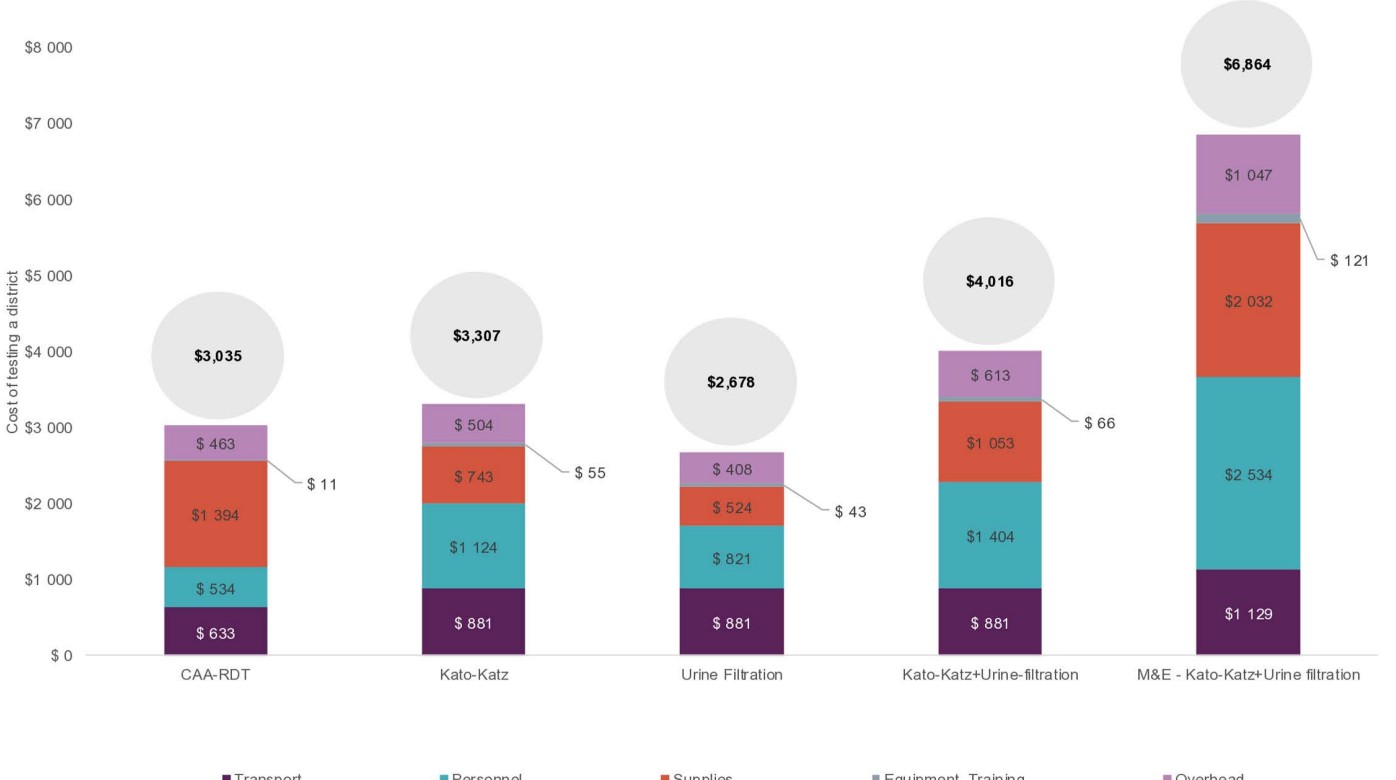

**Fig 1. Total cost of testing a district using traditional sampling: CAA RDT, Kato-Katz, Urine filtration, Kato-Katz and Urine-filtration, and M&E: Kato-Katz and Urine filtration, by cost category.** The total cost corresponding to the y-axis is in the circle. M&E: monitoring and evaluation.

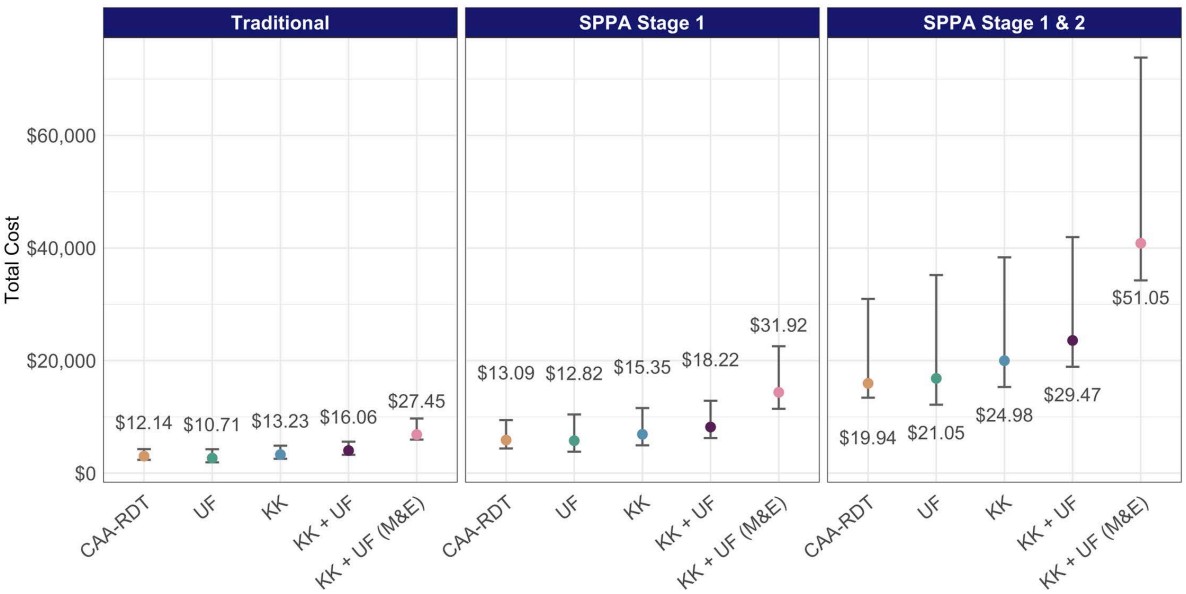

**Fig 2. Total cost to test a district by testing methodology and sampling strategy.** Points represent the expected cost to test a district, while error bars represent minimum and maximum district costs. Labeled costs are the cost per school-age child sampled under the given strategy and test based on the expected district cost. UF: urine filtration, KK: Kato-Katz, M&E: monitoring and evaluation.

was warranted, the cost per SAC was considerably higher than under the traditional sampling methods (64–97%). At the minimum TPP price of $3 per CAA RDT test kit, Kato-Katz became cheaper than the CAA RDT only when staffing costs were halved for traditional sampling (3–12% less), and for SPPA sampling, only when both transport and staffing costs were halved (5–7% less) (Fig A in S1 Text). If the CAA RDT test kit price was decreased to $2.33 from $3, it was always cheaper than Kato-Katz under all sampling and costing scenarios.

## Modelling results

**Test performance.** In the modelled scenarios, the CAA RDT test specifications that could correctly classify at least 80% of sub-districts for appropriate treatment varied by district archetype (Fig 3). When district prevalence was below the 10% prevalence threshold for treatment, a test with high specificity was necessary to correctly classify at least 80% of sub-districts. For districts above the 10% prevalence threshold, the opposite was true, wherein a test with high sensitivity led to a greater proportion of correctly classified sub-districts. This was true for both homogeneous and heterogeneous settings. If test sensitivity was low, correct classification could still be achieved with low test specificity, as false positive and false negative results compensated for one another. However, in high prevalence heterogeneous districts, there was no combination of test specifications that correctly classified at least 80% of sub-districts (Fig 3).

**Comparison of sampling strategies.** Performance of the sampling strategies varied by the district's prevalence distribution, but both followed the same trends with respect to test specifications. When considering variation in test specifications, both sampling strategies tended to overtreat sub-districts in heterogeneous settings and low-prevalence homogeneous settings, but to undertreat in homogeneous settings where the prevalence was 12% (Fig 4). High test specificity (100%) was associated with less overtreatment compared to low specificity (95%) (Fig 4A) but resulted in greater undertreatment in 12% homogeneous settings, attenuated if sensitivity was high (Fig 4B). In general, with a highly specific test (100%), SPPA under- and overtreated less than traditional sampling (except in 8% homogeneous districts). In the sensitivity analysis, SPPA sampling in districts with bimodal prevalence distributions followed the same trends (Fig B in S1 Text).

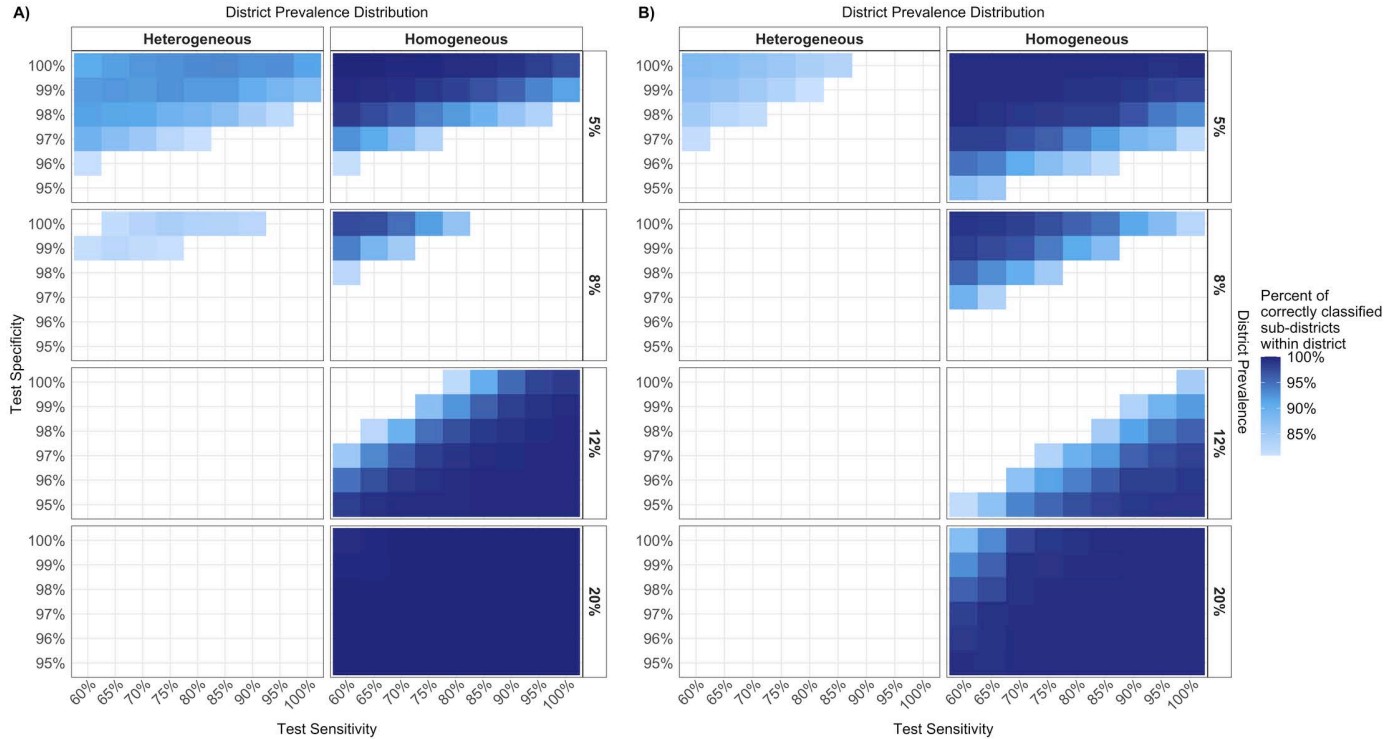

**Fig 3. Test specifications that led to the correct treatment classification for at least 80% of sub-districts by district archetype and sampling strategy: A) SPPA, B) Traditional.**

**Minimum and optimal diagnostic specifications.** The minimum diagnostic specifications that correctly classified at least 80% of sub-districts differed by sampling strategy and prevalence distribution. In the context of the sampling strategies, minimum diagnostic specifications were identified as the specifications with the lowest Youden's J statistic that led to at least 80% of correct sub-district classification, on average, across prevalence and distribution settings. In homogeneous districts, both sampling strategies could maintain a diagnostic sensitivity of 60% but required a minimum specificity of 97% to correctly classify 80% of sub-districts among most district archetypes (Table B in S1 Text). In heterogeneous settings, neither sampling strategy correctly classified 80% of sub-districts within all district archetypes, but when averaged across heterogeneous archetypes, SPPA sampling could achieve this with a sensitivity of 60% and specificity of 98%. When averaged across all district archetypes, SPPA sampling would require 97% specificity (Table B in S1 Text). However, the diagnostic specifications that correctly classified the greatest proportion of sub-districts, on average, across all district archetypes, were a specificity of 100% and a sensitivity of 85% for SPPA sampling (87% correct) or 95% sensitivity for traditional sampling (79% correct).

## Economic analysis

Assuming a test performance of 80% sensitivity and 100% specificity, and averaged across district archetypes, SPPA sampling would correctly treat 87% of sub-districts, while traditional sampling would correctly treat 78% of sub-districts but at a significantly lower cost (Table 2). Whilst the traditional sampling cost per district is constant regardless of the test performance, the test performance influences the probability that SPPA phase 2 sampling is required and, as such, alters the average cost for SPPA sampling. Assuming equal performance across testing technologies, the CAA RDT has a lower cost per outcome (i.e.,

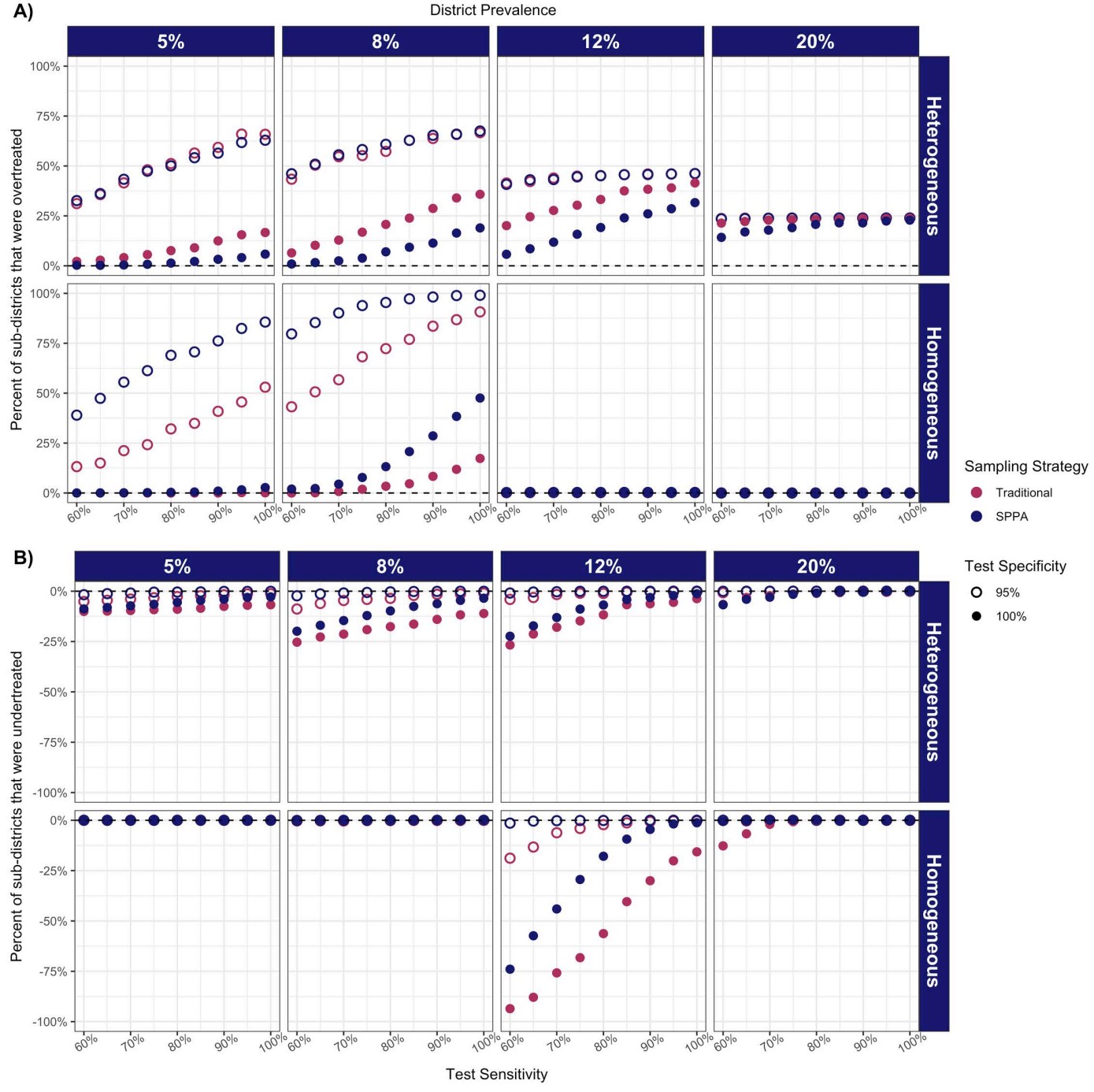

**Fig 4. Percent of sub-districts that were classified for overtreatment (A) or undertreatment (B), by sampling strategy (color: red-traditional, blue-SPPA), test sensitivity, and specificity (shape: hollow-95%, solid-100%).**

cost to correctly classify 10 sub-districts) and is a cost-saving strategy relative to all other test methodologies for SPPA sampling. The CAA RDT has a cost per outcome of $3,891 if traditional sampling is used relative to Kato-Katz of $4,240. The use of urine filtration alone through traditional sampling would be a cost saving strategy compared to the CAA RDT.

**Table 2. Cost and percent of correctly treated sub-districts by sampling strategy and test technology.**

| Test Methodology | Test performance assumptions | Percent sub-districts correctly treated | | SPPA Average Cost (per district) | Traditional cost (per district) | SPPA cost per outcome | Traditional cost per outcome |
|---|---|---|---|---|---|---|---|
| | | SPPA | Traditional | | | | |
| CAA RDT | 80% sensitive, 100% specific | 87% | 78% | $10,921 | $3,035 | $12,553 | $3,891 |
| Urine filtration | | | | $11,285 | $2,678 | $12,971 | $3,433 |
| Kato-Katz | | | | $13,423 | $3,307 | $15,429 | $4,240 |
| Kato Katz and Urine filtration | | | | $15,862 | $4,016 | $18,232 | $5,149 |
| Kato Katz and Urine filtration (M&E) | | | | $27,558 | $6,864 | $31,676 | $8,800 |

## Discussion

The WHO's target to eliminate schistosomiasis as a public health problem has increased demand for praziquantel in endemic countries, particularly with updated guidelines recommending broader population coverage [3]. To avoid unnecessary strain on the global praziquantel supply, new diagnostic and sampling strategies are needed to support more targeted treatment. Our analysis demonstrates that the combination of newer precision mapping strategies (SPPA) with a novel rapid diagnostic test—the CAA RDT—could offer a more efficient and potentially cost-effective approach to precision mapping, as compared to Kato-Katz microscopy. This approach helps reduce both under- and overtreatment of sub-districts compared to traditional district-level sampling.

Despite the CAA RDT's higher per-test cost ($3) compared to microscopy or urine filtration, its operational advantages—reduced labour and transportation requirements—as well as the capacity to detect multiple *Schistosoma* species, make it competitively priced across a range of settings. Particularly in scenarios requiring both stool and urine-based microscopy, or in monitoring and evaluation contexts that involve multi-day sampling, the CAA RDT offers significant savings. Our cost analyses indicate that at the WHO TPP price point of $3, the CAA RDT is cost-saving under most assumptions, especially when SPPA sampling is used. If priced below $2.33, the CAA RDT would consistently outperform microscopy in terms of affordability. This is an important implication for price negotiations in securing tests as no CAA RDT has yet come to market.

Our modelling shows that test performance requirements vary depending on the underlying epidemiology and sampling strategy. In low-prevalence settings (< 10%), high specificity is critical to avoid overtreatment, while high sensitivity becomes important in higher-prevalence areas (> 10%) to ensure adequate treatment. Importantly, our findings suggest that optimal test specifications are not necessarily the highest test specifications possible: a test with 100% specificity and 85% sensitivity led to the greatest proportion of correct treatment classifications under SPPA. These insights refine the criteria laid out in the WHO TPP by emphasizing how performance thresholds interact with implementation—providing more nuanced guidance for test development and evaluation.

As high test specificity was identified as the critical metric for the ideal performance of a SCH diagnostic, Kato-Katz and urine filtration would perform well across sampling strategies and prevalence settings, with an assumed sensitivity of 80% and specificity of 100%. Therefore, the CAA RDT would need to maintain a high specificity to have comparative performance to the standard diagnostic modalities. Kato-Katz sensitivity is dependent on the number of samples collected, slides read, and infection intensity, ranging from 50–100%. Therefore, in low transmission settings where infection intensity is low, Kato-Katz will have low sensitivity, and more sensitive diagnostics will be needed [13,29]. Some studies have shown high sensitivity for CAA detection, even more so than Kato-Katz, urine filtration and the CCA in low transmission settings [13,30–33], and additional performance evaluations of the CAA RDT in the field are still underway. Kato-Katz

is, however, a semi-quantitative diagnostic, in that egg counts indicate infection intensity, which is necessary to measure the elimination of heavy intensity infections and assess elimination in communities. Yet, Kato-Katz has exhibited wide variation in day-to-day egg counts and stool sample consistency, so may also be unreliable in assessing elimination goals, hence the need for two-day sampling for M&E. [34]. It is not yet known if the CAA RDT can also serve as a semi-quantitative diagnostic, but it is believed that CAA concentration in the blood correlates with infection intensity, leaving this a possibility [12,35,36]. Additionally, de Jong et al evaluated a field deployable version of a CAA detection technology that can quantify CAA concentration and found it was comparable to the lab-based technology [37]. So, as communities progress towards elimination, the CAA RDT could be a promising tool to detect infections of low intensity, but more research is needed to understand if CAA concentration can indicate infection intensity and therefore assist in assessing elimination goals.

To the best of our knowledge, this study is the first of its kind to model a CAA RDT for SCH to determine its minimum diagnostic requirements under different sampling strategies and compare its economic outcomes to standard diagnostic techniques. Knowles et al presented an initial step in the development of the SPPA sampling methodology, assessing a two-stage cluster survey design through simulation modelling, and found sampling 15–20 schools per district and 20–30 children per school maximized cost efficiency and minimized undertreatment [38]. In our analysis, both the SPPA and traditional sampling strategies overtreated more than undertreated so undertreatment seems less likely (except in homogeneous districts just above the 10% prevalence threshold). Worrell et al conducted a cost analysis of the POC-CCA and Kato-Katz microscopy in western Kenya, finding comparable per person costs between single Kato-Katz readings and the POC-CCA ($8.75 vs. $9.22, inflated to 2023 USD) [4]. Principal cost drivers included labour and transportation, which concur with our analysis, but our costs were higher, largely driven by supply costs (for example, $3 versus $1.98 for the RDTs), transport costs and double reading of slides for Kato-Katz.

Our modelling approach simulated districts as broadly homogeneous or heterogeneous at the sub-district level, which may be an oversimplification that does not fully capture the distribution of prevalence in the real world. To address this simplification, we simulated district archetypes with bimodal sub-district prevalence distributions. The bimodal results followed similar trends to those observed in heterogeneous districts. This affirms precision mapping strategies are better suited for districts of this composition and could better target treatment to the sub-district level. Similarly, all district archetypes assumed homogeneously distributed prevalence among schools within a single sub-district due to proximity to geographic features (such as water bodies). In a district consisting of 10 sub-districts, traditional sampling would select 0.5 schools per sub-district on average while SPPA sampling would select 1.5 schools, so if school prevalence varies widely within a single sub-district, neither strategy is well equipped to accurately estimate sub-district prevalence. In such regions, when heterogeneity is suspected, SPPA guidelines recommend progressing directly to precision assessment (phase 2), which samples 4 schools per sub-district and may be more adept at estimating sub-district prevalence. In addition, when determining average performance across district archetypes, we artificially assumed equal weighting of district archetypes. Future work could incorporate true prevalence surfaces to accurately reflect average required performance across districts.

This modelling analysis comes with additional important limitations. First, Kato-Katz and urine filtration are considered the gold standard diagnostics for SCH infection yet test sensitivity can vary widely depending on infection intensity. Since we did not model infection intensity in relation to district prevalence in this analysis, it was difficult to directly compare CAA RDT performance to microscopy. We generously assumed Kato-Katz and urine filtration had a performance that was 80% sensitive and 100% specific. However, in the absence of a gold-standard test against which to compare it, these specifications could be much lower. Second, we did not include the costs of praziquantel treatment in our economic analysis, which could have implications for the desired sampling strategy with respect to overtreatment. Third, we have ignored efficiencies that may arise from the integration of SCH and soil-transmitted helminth mapping surveys and, in areas where there is co-infection, microscopy would still be needed alongside the CAA RDT. The overall prevalence of co-infection of SCH

and soil-transmitted helminths was estimated to be 19.1% in Tanzania [39]. In areas with high co-infection, the continued use of Kato-Katz may be more cost-effective. Fourth, the *WHO Guideline on Control and Elimination of Human Schistosomiasis* (2022) states the treatment threshold of 10% was determined based on Kato-Katz microscopy performance and may differ for tests such as the CCA and CAA [3]. Therefore, the final specifications of the CAA may require a treatment threshold adjustment, while we maintained a 10% threshold in this analysis. Finally, we did not consider an elimination perspective in this analysis, wherein overtreatment in some sub-districts could be of benefit even if it leads to a decrease in the efficiency of resource allocation. Overall, our approach is easily generalisable to other settings, based on whether they are low, moderate, or high prevalence settings. Equally, the sensitivity analysis around staff and transport costs allows for comparison across geographies.

## Conclusion

As SCH control programmes progress towards elimination, the disease is likely to become more focal, resulting in a heterogeneous distribution of prevalence. This in turn will require precision-based mapping to more efficiently target and administer preventive chemotherapy for communities in need. This modelling analysis indicates that a CAA RDT with a high specificity will be a valuable cost-competitive diagnostic tool to be used in SCH prevalence mapping, particularly under precision mapping strategies. Research remains to evaluate the performance of the CAA RDT in the field under specific sampling strategies, as well as its ability to estimate infection intensity, to determine if it could replace microscopy in SCH elimination programmes.

## Supporting information

**S1 Text.**   Additional costing methodology and sensitivity analyses. **Fig A.** Percentage cost difference between Kato-Katz and a CAA RDT testing strategy with varying staff costs, and transport costs. **Fig B.** Percent of sub-districts that were classified for overtreatment when average prevalence was < 10% (A) or undertreatment when average prevalence was > 10% (B), by sampling strategy, test sensitivity, and specificity, in three districts with bimodal prevalence distributions. **Table A.** Cost categories and resource use and prices. **Table B.** Minimum sensitivity and specificity to achieve greater than or equal to 80% correct sub-district treatment, by district archetype and sampling strategy.
(DOCX)

## Acknowledgments

Sammy Njenga and Hilman Ugangu (KEMRI), Fiona Fleming (Unlimit Health, London, UK), Rachel Pullan (LSHTM).

## Author contributions

**Conceptualization:** Sarah Girdwood, Shaukat Khan, Sarah Hingel.

**Formal analysis:** Joshua M. Chevalier, Kyra H. Grantz, Sarah Girdwood.

**Funding acquisition:** Sarah Hingel.

**Methodology:** Joshua M. Chevalier, Kyra H. Grantz.

**Supervision:** Shaukat Khan, Sarah Hingel.

**Visualization:** Joshua M. Chevalier, Kyra H. Grantz.

**Writing – original draft:** Joshua M. Chevalier, Kyra H. Grantz, Sarah Girdwood.

**Writing – review & editing:** Joshua M. Chevalier, Kyra H. Grantz, Sarah Girdwood, Stella Kepha, Thierry Ramos, Brooke E. Nichols, Shaukat Khan, Sarah Hingel.

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
