## [Decision Letter · Decision Letter 0]

13 Jan 2025

PNTD-D-24-01659

Evaluating the impact and cost of a new rapid diagnostic test for school-based prevalence mapping and monitoring and evaluation surveys of Schistosomiasis: a modelling study

Dear Dr. Khan,

Thank you for submitting your manuscript to PLOS Neglected Tropical Diseases. After careful consideration, we feel that it has merit but does not fully meet PLOS Neglected Tropical Diseases's publication criteria as it currently stands. Therefore, we invite you to submit a revised version of the manuscript that addresses the points raised during the review process.

Please submit your revised manuscript within 60 days Mar 14 2025 11:59PM. If you will need more time than this to complete your revisions, please reply to this message or contact the journal office at plosntds@plos.org. Please include the following items when submitting your revised manuscript:

We look forward to receiving your revised manuscript.

Kind regards,

Bruce A. Rosa

Academic Editor

Justin Remais

Section Editor

Shaden Kamhawi

co-Editor-in-Chief

Paul Brindley

co-Editor-in-Chief

**Additional Editor Comments :**

The reviewers have an overall positive view of the manuscript, but have expressed several specific concerns in regards to adding details about the methodology. Please ensure that the revised manuscript contains substantially more methodology details to address these comments, in addition to addressing the other provided feedback.

**Journal Requirements:**

At this stage, the following Authors/Authors require contributions: Joshua Michael Chevalier, Kyra H. Grantz, Sarah Girdwood, Stella Kepha, Thierry Ramos, Brooke E. Nichols, Shaukat Khan, and Sarah Hingel. Please ensure that the full contributions of each author are acknowledged in the "Add/Edit/Remove Authors" section of our submission form.

2) Please insert an Ethics Statement at the beginning of your Methods section, under a subheading 'Ethics Statement'. It must include:

i) The full name(s) of the Institutional Review Board(s) or Ethics Committee(s)

ii) The approval number(s), or a statement that approval was granted by the named board(s)

iii) A statement that formal consent was obtained (must state whether verbal/written) OR the reason consent was not obtained (e.g. anonymity). NOTE: If child participants, the statement must declare that formal consent was obtained from the parent/guardian.].

4) When completing the data availability statement of the submission form, you indicated that you will make your data available on acceptance. We strongly recommend all authors decide on a data sharing plan before acceptance, as the process can be lengthy and hold up publication timelines. Please note that, though access restrictions are acceptable now, your entire data will need to be made freely accessible if your manuscript is accepted for publication. This policy applies to all data except where public deposition would breach compliance with the protocol approved by your research ethics board. If you are unable to adhere to our open data policy, please kindly revise your statement to explain your reasoning and we will seek the editor's input on an exemption. Please be assured that, once you have provided your new statement, the assessment of your exemption will not hold up the peer review process.

**Comments to the Authors:**

**Please note that one of the reviews is uploaded as an attachment.**

**Reviewers' Comments:**

Reviewer's Responses to Questions

**Key Review Criteria Required for Acceptance?**

**Methods**

-Are the objectives of the study clearly articulated with a clear testable hypothesis stated?

-Is the study design appropriate to address the stated objectives?

-Is the population clearly described and appropriate for the hypothesis being tested?

-Is the sample size sufficient to ensure adequate power to address the hypothesis being tested?

-Were correct statistical analysis used to support conclusions?

-Are there concerns about ethical or regulatory requirements being met?

Reviewer #1: Why repeat the Objective in the methods?

Line 140-142. No detailed CAA RDT procedure has been provided.

See Ref: https://doi.org/10.3389/fpara.2024.1460331

No information on the geographic distribution of targeted districts is provided. What’s the sizes of these districts? Can results from 5 schools (50 SAC per school) represent the prevalence of the district?

Lines 166-7, Why sites are selected purposively, not randomly, meaning sites with a higher potential for SCH (i.e. proximity to water bodies or known sites of transmission) are selected preferentially.

Line 167-8, As a proxy for purposive sampling, we used a sampling weight proportional to school prevalence. Could you provide more details on this?

Line 172, what’s the standard for a homogeneous/heterogeneous distribution in a district?

Line 173-4, what does it mean ‘systematically selected per district to ensure equal coverage of sites across sub-districts’? ‘30 SAC are sampled per school’: this sample size is pretty low.

Lines 177-8, What the total population size in each of sub-district? Can results from 4 schools (20 SAC per school) represent the prevalence of the sub-district? How many sub-districts are included here in this study? Again, why four schools are purposively selected from each sub-district?

Line 185, what the model is used exactly here? Is there an algorithm?

Line 190-1, which data are used exactly here. Four references, which were published in different years, re were cited here. Can the prevalence data from these references be merged together?

Reviewer #2: (No Response)

Reviewer #3: The methods section is clear and reads very well. Only minor clarifications that could potentially improve the section:

Cost data – You mention Resource use was determined through interviews. Did you use budgeted expenses or executed expenses in your analysis?

Would be good to understand how did you build your district archetypes. Any underlying assumption, previous studies or simply defined by you?

You mention that prevalence studies conducted in Kenya informed the prevalence you defined for districts. Could you be more specific and detail how it informed? Did you simulate district prevalence that are similar to those found in Kenya?

**Results**

-Does the analysis presented match the analysis plan?

-Are the results clearly and completely presented?

-Are the figures (Tables, Images) of sufficient quality for clarity?

Reviewer #1: Lines 242-49

Which methods were used to obtain these results? How many sub-districts (n=?) How many districts were tested? Are there results based on medolling?

Reviewer #2: (No Response)

Reviewer #3: NOthing to add on this one. Clear and well presented.

**Conclusions**

-Are the conclusions supported by the data presented?

-Are the limitations of analysis clearly described?

-Do the authors discuss how these data can be helpful to advance our understanding of the topic under study?

-Is public health relevance addressed?

Reviewer #1: This modelling analysis comes with many important limitations, such as oversimplification. Thus the conclusion drawn from the modelling analysis may be far from accuracy and reliability.

Reviewer #2: (No Response)

Reviewer #3: Discussion

The first three paragraph broadly repeat the results section. While always a good practice to revisit main results, it looks a bit excessive, particularly considering the detail and objectiveness of the results section. I would advice reducing condensing these two paragraphs to focus the discussion more on the broader comparison with other studies, the usefulness of results and limitations of the study.

Line 377 – it also depends on quality and capacity of diagnosis of microscopists. Would be good to expand your discussion on this by adding references but also highlighting how this test somehow removes the effect of the quality/experience of the microscopist

It is not clear if this new test provides semi quantitative results in any shape or form. When you mention the value for impact assessments, it should be clear that reduction elimination of SCH as a public health problem still depends on quantitative measures as the indicator is measured through the prevalence of heavy infections, which are based on egg counts, and not overall prevalence. My suggestion here is to clarify that this tool can still be a plus for measuring prevalence and target treatments but has its limitation to assess elimination achievements. I understand the WHO guidance for attributing EPHP may be an issue but it is the criteria we have. If the new test provides some sort of quantitative approach. This idea is stated in Background and then again in Conclusions where you state progress towards elimination but then mention SCH prevalence mapping. It shoudl be clear that these tests, as they are, have their own limitations to assess SCH elimination achhievements

**Editorial and Data Presentation Modifications?**

Reviewer #1: (No Response)

Reviewer #2: (No Response)

Reviewer #3: (No Response)

**Summary and General Comments**

Reviewer #1: The study by Chevalier aims to evaluate the impact and cost of a new rapid diagnostic test, POC-CAA, for school-based prevalence mapping and monitoring and evaluation surveys of schistosomiasis using a modelling analysis. While the topic is of interesting, the manuscript suffers from some weaknesses. The methods section is still very elusive, with lots of key information missing. The rational for the modelling analysis is unclear. In addition, the modelling has lots of limitations as stated in the discussion section such as oversimplification, indicating that the conclusions drawn from a simplified modelling may be far from accuracy and reliability. In the title, it states that this is a modelling study. However, no mathematical model has been mentioned in the Abstract and Introduction sections.

Abstract section

1. Background: The last sentence sounds a conclusion. A prototype circulating anodic antigen rapid diagnostic test (CAA RDT) could replace the standard tools for mapping (Kato-Katz and urine-filtration) and better support precision-mapping and thus more efficient drug distribution.

2. Methodology: The section is descriptive, without any detailed methods provided, e.g., sample size? The first sentence is very long and difficult to follow.

3. Results: some sentences are like conclusions: In settings with both S. mansoni and S. haematobium, a CAA RDT with comparable performance to Kato-Katz/urine filtration would be cost-saving. Further,

the use of a CAA RDT for M&E would be the most cost-effective strategy as compared to two consecutive day sampling required with Kato-Katz.

Author Summary

Line 69, Using a mathematical model. Which model? Why the model used here can achieve the purpose of the study? What’s the rational of the modelling?

Introduction section

Line 90, Can the author provide more details about: ‘often characterized by heterogeneous geographical distribution’? What’s the geographical level? District? Sub-district? Village?

Reviewer #2: (No Response)

Reviewer #3: This is an interesting paper that measures several key components essential to measure progress of SCH programs towards elimination. New diagnostic tools are key to keep progress as well as assessing the best sampling strategies and comparing the most cost effective mapping approaches. Well done in doing it as it provides a very interesting modelling exercise with clear data and values that can be useful for decision making, particularly to decide on the costs of traditional vs oversampling approaches.

The main weakness of the paper, in my opinion, is it lack of translation into practical public health measures. The discussions mostly repeats the results and does not provide a broader discussion about the implications of these results: what should program managers do with these results, what further research should be done to validate these results with real field data? Any recommendations for WHO to adjust sampling procedures and/or diagnostic approaches? The paper misses the research translation element that is key to put research into practice and would certainly improve if such element is added.

PLOS authors have the option to publish the peer review history of their article (what does this mean? ). If published, this will include your full peer review and any attached files.

**Do you want your identity to be public for this peer review?** For information about this choice, including consent withdrawal, please see our Privacy Policy .

Reviewer #1: No

Reviewer #2: No

Reviewer #3: **Yes: ** Sergio Castela Lopes

**Figure resubmission:**
---

## [Decision Letter · Decision Letter 1]

17 Mar 2025

PNTD-D-24-01659R1The impact and cost of a new rapid diagnostic test for school-based prevalence mapping and monitoring and evaluation surveys of Schistosomiasis: a modelling studyPLOS Neglected Tropical Diseases Dear Dr. Khan, Thank you for submitting your manuscript to PLOS Neglected Tropical Diseases. After careful consideration, we feel that it has merit but does not fully meet PLOS Neglected Tropical Diseases's publication criteria as it currently stands. Therefore, we invite you to submit a revised version of the manuscript that addresses the points raised during the review process. Please submit your revised manuscript within 30 days Apr 16 2025 11:59PM. If you will need more time than this to complete your revisions, please reply to this message or contact the journal office at plosntds@plos.org. Please include the following items when submitting your revised manuscript: * A rebuttal letter that responds to each point raised by the editor and reviewer(s). You should upload this letter as a separate file labeled 'Response to Reviewers '. This file does not need to include responses to any formatting updates and technical items listed in the 'Journal Requirements' section below.* A marked-up copy of your manuscript that highlights changes made to the original version. You should upload this as a separate file labeled 'Revised Manuscript with Track Changes '.* An unmarked version of your revised paper without tracked changes. You should upload this as a separate file labeled 'Manuscript '. If you would like to make changes to your financial disclosure, competing interests statement, or data availability statement, please make these updates within the submission form at the time of resubmission. Guidelines for resubmitting your figure files are available below the reviewer comments at the end of this letter.

We look forward to receiving your revised manuscript.

Kind regards,

Bruce A. Rosa

Academic EditorPLOS Neglected Tropical Diseases Justin RemaisSection EditorPLOS Neglected Tropical Diseases

Shaden Kamhawi

co-Editor-in-Chief

Paul Brindley

co-Editor-in-Chief

**Additional Editor Comments :** With the revised manuscript, the authors have satisfied most of the reviewer comments. However, reviewer 1 has some additional suggestions for revisions. Please carefully consider these additional points with one more round of revision.

**Reviewers' comments:**

Reviewer's Responses to Questions

**Key Review Criteria Required for Acceptance?**

**Methods**

-Are the objectives of the study clearly articulated with a clear testable hypothesis stated?

-Is the study design appropriate to address the stated objectives?

-Is the population clearly described and appropriate for the hypothesis being tested?

-Is the sample size sufficient to ensure adequate power to address the hypothesis being tested?

-Were correct statistical analysis used to support conclusions?

-Are there concerns about ethical or regulatory requirements being met?

Reviewer #1: No subtitle for the first paragraph. Still it should not belong to this section.

Ln 131, Ethics statement

Ln 135, Diagnostic technologies

Ln 155, what does KEMRI stand for?

Ln 169, Sampling strategies

Ln 270-3, We simulated eight different district archetypes, each consisting of 10 sub-districts with 50 schools per sub-district, with varying prevalence distributions. District size, with respect to number of sub-districts and primary schools per sub-district, was parameterized to demographic data from Kenya [15–17].

However, the rational for this simulation is unclear in the main text but in the response to my previous comment #3, i.e., the district size in this analysis would correspond to a moderately sized district of one-million people, divided into 10 sub-districts of roughly 100,000 people each (based on 2019 census data). We assume 20% of the population (20,000/sub-district) is school-aged children, divided into schools of an average size of 370 (based on the Kenya Demographic and Health Survey of 2022 and the Basic Education Statistical Booklet of 2020), rounded to 50 schools per sub-district. These details should be included in the main text to increase the logicality, readability, and reproducibility.

In addition, to ensure transparency and reproducibility of the study, the authors should mention that the model has been publicly available on Figshare (DOI: https://doi.org/10.6084/m9.figshare.27325005.v1.) in the main text or attach the model as supplementary data, not just in the response to reviewers’ concerns.

Table 1 should be cited after the sentence: District archetypes included a low prevalence district (average prevalence 5%); moderate prevalence districts near the 10% prevalence threshold (8% and 12%), and a high prevalence district (20%), all with either a homogeneous or heterogeneous distribution of sub-district prevalence.

Ln 206-7: Prevalence among children at each school within a sub-district is assumed to be homogeneously distributed (standard deviation of 0.75% around the mean sub-district prevalence).

How about the prevalence among children at each school within a sub-district is assumed to be heterogeneously distributed?

**Results**

-Does the analysis presented match the analysis plan?

-Are the results clearly and completely presented?

-Are the figures (Tables, Images) of sufficient quality for clarity?

Reviewer #1: Ln 228, Difficult to follow the sentence:

The cost of testing a district using traditional sampling with the CAA RDT was on par with the cost of testing using ?(Fig 1)

Fig 1. It is difficult to observe the number in dark purple bar.

Ln 242-3, for phase 1 was only 8–12% more expensive per SAC than traditional sampling across both diagnostics, but if phase 2 was warranted, the cost per SAC was considerably higher than under the traditional sampling method (113–139%).

How to get these percentages? Could you check again? Also check the percentages (3-12% less) in Ln 245, and (5-7% less) in Ln 246.

Ln 253, Modelling results

Ln 254, Test performance

Fig 4 the symbol legend is a bit confusing

Ln 303, $3,891 if traditional sampling is used relative to Kato-Katz of $4,240.

**Conclusions**

-Are the conclusions supported by the data presented?

-Are the limitations of analysis clearly described?

-Do the authors discuss how these data can be helpful to advance our understanding of the topic under study?

-Is public health relevance addressed?

Reviewer #1: The limitations of analysis are clearly described.

Ln 620, We have shown that a CAA RDT with a high specificity will be

Maybe mention the model here to void confusion.

**Editorial and Data Presentation Modifications?**

Reviewer #1: (No Response)

**Summary and General Comments**

Reviewer #1: The current revision improves the quality of the manuscript, but still there are lots of issues need to be addressed.

Title: The impact and cost of a new rapid diagnostic test for school-based prevalence mapping and monitoring and evaluation surveys of schistosomiasis: a modelling study

Ln 38: In endemic communities where the prevalence of schistosomiasis

Introduction:

References are missing for a number of sentences in the Introduction section, such as:

A circulating cathodic antigen (CCA) rapid diagnostic test (RDT) has been commercially available as a point-of-care test since 2008, however, despite being easier to use and less labor intensive than Kato-Katz, it only works well for detecting S. mansoni infections and its sensitivity decreases with decreasing infection intensity.

A new RDT, (still under-development), can detect circulating anodic antigen (CAA) in an infected host’s blood, as a marker for active infection for S. mansoni, S. japonicum and S. haematobium.

Based on early evaluations, the CAA RDT prototype has the potential to reliably measure prevalence in SAC as well as in adults.

Ln 311, As schistosomiasis

Ln 315, a novel rapid diagnostic test in the CAA RDT?

Ln 323, the cost of testing a district using Kato-Katz ($3,035 v. $3,307,

Ln 324, the CAA RDT at $2,678 per

Ln 327, all three Schistosoma (italic)

Ln 331-2, Which result supports this statement: In a sensitivity analysis, staffing and transport costs were halved to approximate costs that would be applicable to smaller regions such as the Philippines where S. japonicum is present, noting that lots of parameters used in the model were based on the Kenyan data. In addition, the transport costs would not be cheap in the Philippines as goods were transported by airplane between islands. Why the regions in the Philippines are smaller?

Ln 369, Additionally, in [35]

Is it the journal’s style to cite a reference. The same issue applies to Ln 378 and Ln 383.

The first three paragraphs in the discussion section remain repetitive to background and result sections.

PLOS authors have the option to publish the peer review history of their article (what does this mean? ). If published, this will include your full peer review and any attached files.

**Do you want your identity to be public for this peer review?** For information about this choice, including consent withdrawal, please see our Privacy Policy .

Reviewer #1: No

**Figure resubmission:**
---

## [Editor Report · Decision Letter 2]

17 Apr 2025

Dear Dr. Khan,

We are pleased to inform you that your manuscript 'The impact and cost of a new rapid diagnostic test for school-based prevalence mapping and monitoring and evaluation surveys of Schistosomiasis: a modelling study' has been provisionally accepted for publication in PLOS Neglected Tropical Diseases.

Best regards,

Bruce A. Rosa

Academic Editor

Justin Remais

Section Editor

Shaden Kamhawi

co-Editor-in-Chief

Paul Brindley

co-Editor-in-Chief

---

## [Editor Report · Acceptance letter]

Dear Dr. Khan,

We are delighted to inform you that your manuscript, "The impact and cost of a new rapid diagnostic test for school-based prevalence mapping and monitoring and evaluation surveys of schistosomiasis: a modelling study," has been formally accepted for publication in PLOS Neglected Tropical Diseases.

Best regards,

Shaden Kamhawi

co-Editor-in-Chief

Paul Brindley

co-Editor-in-Chief
